# Transmission Cycle of Tick-Borne Infections and Co-Infections, Animal Models and Diseases

**DOI:** 10.3390/pathogens11111309

**Published:** 2022-11-08

**Authors:** Sandra C. Rocha, Clara Vásquez Velásquez, Ahmed Aquib, Aya Al-Nazal, Nikhat Parveen

**Affiliations:** Department of Microbiology, Biochemistry and Molecular Genetics, Rutgers New Jersey Medical School, Newark, NJ 07103, USA

**Keywords:** co-infection, reservoir host, animal-model, tick-borne disease, tick, prevalence

## Abstract

Tick-borne pathogens such as species of *Borrelia*, *Babesia*, *Anaplasma*, *Rickettsia*, and *Ehrlichia* are widespread in the United States and Europe among wildlife, in passerines as well as in domestic and farm animals. Transmission of these pathogens occurs by infected ticks during their blood meal, carnivorism, and through animal bites in wildlife, whereas humans can become infected either by an infected tick bite, through blood transfusion and in some cases, congenitally. The reservoir hosts play an important role in maintaining pathogens in nature and facilitate transmission of individual pathogens or of multiple pathogens simultaneously to humans through ticks. Tick-borne co-infections were first reported in the 1980s in white-footed mice, the most prominent reservoir host for causative organisms in the United States, and they are becoming a major concern for public health now. Various animal infection models have been used extensively to better understand pathogenesis of tick-borne pathogens and to reveal the interaction among pathogens co-existing in the same host. In this review, we focus on the prevalence of these pathogens in different reservoir hosts, animal models used to investigate their pathogenesis and host responses they trigger to understand diseases in humans. We also documented the prevalence of these pathogens as correlating with the infected ticks’ surveillance studies. The association of tick-borne co-infections with other topics such as pathogens virulence factors, host immune responses as they relate to diseases severity, identification of vaccine candidates, and disease economic impact are also briefly addressed here.

## 1. Introduction

Tick-borne diseases (TBDs) are caused by various pathogens, including bacteria, viruses, protozoa, and helminths, and they are becoming a significant public health concern worldwide [1]. Moreover, ticks could be co-infected with different combinations of pathogens to facilitate their simultaneous transmission to humans and animals. Wildlife, usually small animals, are primary reservoir hosts in the natural cycle of tick-borne pathogens (TBPs) and are responsible for transmitting these microbes to humans and farm and companion animals through ticks. Ticks are growing in terms of abundance (increasing in number with higher frequency of detection) and are expanding in new geographical areas despite various strategies employed for controlling tick populations. Climate change and longer periods of warmer temperature are favorable for their development and appear to facilitate the expansion of tick populations. Significant efforts are also undertaken for effective disease diagnostics development and timely treatment to fight the increased emerging cases of TBDs worldwide [2,3]. 

Pathogens causing TBDs with greater public health interest belong to the bacterial species: *Borrelia burgdorferi* sensu lato (s.l.) and *Borrelia mayonii* (both causative agents of Lyme disease), *Borrelia miyamotoi* (relapsing fever), *Rickettsia rickettsii* (Rocky Mountain spotted fever), *Anaplasma phagocytophilum* (anaplasmosis) and *Ehrlichia* spp. (ehrlichiosis). In addition, protozoa belonging to *Babesia* (henceforth *Ba.* spp. that causes babesiosis), *Theileria* and *Hepatozoon*, and viruses belonging to Flaviviridae: tick-borne encephalitis virus (TBEV) and Crimean-Congo hemorrhagic fever also affect a significant number of individuals [3,4]. In 2019, the Centers for Disease Control and Prevention (CDC) reported 50,865 cases of TBDs in the United States; of these, the four most prevalent illnesses are represented by Lyme disease (LD) (34,945 cases), anaplasmosis/ehrlichiosis (7976 cases), spotted fever rickettsiosis (5207 cases) and babesiosis (2420 cases) [5]. An estimated 476,000 individuals are treated for LD in the United States annually, with less than 10% of case numbers being reported [6,7]. In the United States, the most frequent co-infection is LD and babesiosis [8,9], whereas in Europe, tick-borne encephalitis (TBE) caused by the virus is a common TBD that co-exists with LD [8,10,11,12].

Tick-borne co-infections (TBCI) are acquired by ticks during blood meal from a co-infected host or by sequential feeding during different developmental stages on different infected hosts [13]. As a result, hosts may either become co-infected through transmission by bite of a single tick harboring multiple pathogens or during bloodmeal of multiple ticks, which are infected by different pathogens, feeding simultaneously or sequentially. Co-existence of multiple pathogens was first reported in the 1970s by observation of arboviruses TBE and Uukuniemi (UUK) circulating in Norwegian *Ixodes ricinus* ticks [14]. In the United States, the first co-infection in rodents was reported with *Ba. microti* and *Borrelia burgdorferi* in 1984 [15], and human cases of Babesiosis–LD together were reported in the same decade in New York, Massachusetts, and Wisconsin [9,16,17,18]. One of the first human cases of TBEV and *B. burgdorferi* co-infection was reported in Germany in 1986 [19]. 

Both wild and domestic animals play an important role in the transmission cycle of TBPs to human hosts, which has become a major concern for public health. Companion animals such as dogs and cats that spend time outdoors are particularly at risk of contracting TBDs, mostly those caused by *A. phagocytophilum*, *Ba. microti*, *B. miyamotoi*, and *B. burgdorferi* [20,21]. Farm animals such as cattle, horses, sheep, and goats are constantly exposed to ticks and are also at risk of TBCI [22,23,24,25,26]. As reported previously, wildlife has tremendous potential of serving as a source of TBDs for domestic animals and humans [2,25,27,28]. Due to the close interaction between humans and various animals, the data integration from veterinary, clinical, and wildlife surveillance reports is extremely important for a better understanding of TBCI [2]. 

In this review, we describe TBCI, focusing on their prevalence in different reservoir animals. We also summarize reports on: (1) the determination of the prevalence of TBCI in different wild animals and their correlation with surveillance studies of infected ticks; (2) the major virulence factors identified in pathogens causing diseases; (3) the impact of co-infection on host immune response and disease severity; (4) the use of animal models for infection, co-infection, and pathogenesis studies; (5) the evaluation of various vaccine candidates; and (6) the economic impact of these TBDs. We also briefly cover the impact of TBDs/TBCI on humans.

## 2. Ticks as Vectors of Transmission of Co-Infections

Ticks are cosmopolitan ectoparasites that feed on the blood of terrestrial vertebrates, including mammals, birds, amphibians, and reptiles. The threat of their widespread presence is of great concern due to their capacity to harbor pathogens that cause serious diseases in humans and animals. Among the two major families of ticks, Argasidae (soft ticks) are prevalent in tropics and subtropics, whereas Ixodidae (hard ticks) have a more considerable prevalence in temperate regions [29]. Soft ticks are of comparably lesser medical and veterinary importance, as most species feed on wildlife, especially bats [30]; however, soft ticks of the genus Ornithodoros are epidemiologically consequential as vectors of diseases such as Argasidae-borne relapsing fever (ABRF) and African swine fever (ASF) [31,32]. Hard ticks are responsible for transmitting most TBPs, including bacteria, viruses, and protozoans. Among hard ticks, *Ixodes* species are the most prominent vectors worldwide and are responsible for transmitting pathogenic species of *Borrelia*, *Anaplasma*, *Babesia*, and *Ehrlichia*, as well as TBEV [33]. The *Ixodes* spp. have a distinguishable geographical distribution, and *I. scapularis* and *I. pacificus* are prevalent along North America’s eastern and western coasts, respectively. Most parts of Europe, Western Asia, and North Africa have established populations of *I. ricinus* and *I. persulcatus*, which prominently occur in Southern Russia [34]. Alarmingly, the geographical distribution of tick species is constantly expanding due to changes in climate, habitat, and anthropogenic activities [35]. 

TBPs are maintained in enzootic cycles in reservoir hosts that are primarily small mammals in cases of epidemiologically important pathogens. These pathogens are amplified by passing from reservoir hosts to ticks and vice versa [36]. The feeding period of hard ticks, unlike soft ticks, is longer and varies from 3 to 12 days such that the transmission of pathogens requires prolonged contact of ticks with hosts for efficient transmission [37,38]. In the Eastern US, the white-footed mouse, a reservoir host, is often infected by both *B. burgdorferi* and *Ba. microti* [39]. In the nymphal tick population from the same region, *B. burgdorferi* infections are twice as frequent as *Ba. microti*. A similar disproportion of infection was also observed in a laboratory experiment suggesting that the efficient transmission from host to tick plays a vital role in the zoonotic expansion of *B. burgdorferi* [40]. A complex interplay between various factors related to ticks, reservoir hosts, and microbes themselves determines the prevalence of pathogens together, and their potency to cause co-infections also depends upon tick and host density in a particular region [41]. The most critical factors responsible for establishing co-infections are the feeding pattern of ticks and duration of infectivity in the vector while the immune mechanisms of hosts and ticks determine the persistence of co-infections [42,43]. For example, a previous analysis of pathogenic interactions suggests that co-infection with *B. burgdorferi* and *Ba. microti* enhances the survival of the latter, and as a result, Lyme spirochetes also promote transmission of *Ba. microti* in the enzootic cycle in endemic regions for tick-borne infections [34].

Although the density of the reservoir host has a strong positive effect on the density of nymphs, its effect on the density of infected nymphs varies for different pathogens [44,45]. This variability can be partially explained by the preferred host responsible for transmission of the pathogen. For example, the density of nymphs infected by *B. afzelii* and *Neoehrlichia mikurensis*, which primarily rely on rodents for amplification, has a strong positive association with rodent density, whereas the bird feeding associated *B. garinii* and the transovarially enhanced *Spiroplasma ixodetis* lack a strong positive association between rodent density and density of infected nymphs [45]. In the strains of *Anaplasma marginale*, tick-pathogen interaction has been reported to play a significant role in transmission efficiency. For example, the St. Maries strain was found to have a ten-fold higher pathogen load in the salivary gland than the low-efficiency vaccine strain of this bacterium [46]. In addition to transmission efficiency, the vector infection threshold, i.e., the lowest number of infectious units capable of causing an infection to about 1 to 5% of the vector population, is a factor that is critical in determining the zoonotic prevalence of various pathogens [47]. 

Transmission dynamics and thereby pathogen prevalence can also be affected either by negative or positive interaction between pathogens within ticks. Furthermore, a change in behavior or the ability to survive adverse conditions conferred by one pathogen could lead to changes in the transmission dynamics of co-existing pathogens. For example, increased tolerance for thermohydrometric conditions has been observed in *B. afzelii* infection of *I. ricinus* ticks [48]. Likewise, *A. phagocytophilum* presence can hinder molting of *I. scapularis* larvae into nymphs and thereby tick survival [49]. The evidence of a strong positive interaction between pathogens within ticks is not yet available. Several pathogens have been found to exist simultaneously more often in ticks than can be expected based upon chance alone [25,28,50], such that this co-existence could generally be attributed to ticks feeding on co-infected hosts (Figure 1). Supporting this premise, studies in murine models have shown that immune modulation during co-infection, such as of *B. burgdorferi* and *A. phagocytophilum*, resulted in increased pathogens burden and enhanced transmission from host to ticks [51,52].

Ixodid ticks have been examined by researchers for co-infection of epidemiologically important pathogens, especially *B. burgdorferi*, *A. phagocytophilum*, and *Ba. microti*. Overall, ~3–20% of mixed infections have been observed with ranges primarily depending on geographic area and the number of pathogens screened. In Europe, the proportion of mixed infections in adult ticks was reported to be 20.2% in France [13], 12.3% in Denmark [53], 6.7% in Switzerland [54], 6.3% in Latvia [55] and 3.2% in Luxembourg [56]. Adult ticks are more likely to be co-infected than nymphs because they have had additional blood feeding. Correspondingly, in Denmark, the same study reported that only 3.5% of co-infection was present in nymphs [53]. In the United States, co-infections are most common in the northeast, where 11.2% of adults and 7.6% of nymphs were reported to be co-infected mostly by *B. burgdorferi* with either *A. phagocytophilum* or *Ba. microti* [57]. 

## 3. Wildlife as a Reservoir Host for TBCI

In nature, TBPs have been reported in different species ranging from small to large animals worldwide (Table 1) as listed for the US [70], Canada [72], South America [65,74], Europe [28,63,68,69,75], Australia [76] Africa [77] and China [78]. Despite tick blood meal being the most common transmission mode of TBPs, oral engulfment of infected ticks by animals, carnivorism, animal bites, and transplacental routes also occur to cause wildlife infection [2]. Studies have shown that animals in their natural habitats can co-harbor and co-transmit two or more pathogens to their attached ticks [70]. Furthermore, pathogen interaction can also drive infection risk in wildlife [79]. 

### 3.1. Small Mammalian Species

Small mammals are the most important reservoirs for TBPs, mostly due to their rapid reproduction and easy adaptation to hostile habitats and closer to the land (and ticks) existence (Figure 2). The white-footed mouse (*Peromyscus leucopus*) is the most studied rodent host for TBPs in the United States, such as for *A. phagocytophilum*, *Ba. microti*, and *B. burgdorferi*, due to its higher competence in maintaining and transmitting pathogens to ticks [13,80,81]. In an early study, TBPs were assessed in white-footed mice and meadow voles (*Microtus pennsylvanicus*) captured on two islands in Narragansett Bay in the northeastern US. The authors reported that 42.8% of rodents were infected with *B. burgdorferi*, 21.4% with *Ba. microti* and 35.7% were co-infected [15]. In New York, the most prevalent TBCIs observed were *B. burgdorferi*-*Ba. microti* and *B. burgdorferi*-*A. phagocytophilum* in fed *I. scapularis* ticks collected from white-footed mice and eastern chipmunks (*Tamias striatus*) [70]. Co-infections with *B. afzelii*-*N. mikurensis*, and *B. miyamotoi*-*N. mikurensis* were reported in tissue samples of rodents from urban Romania, and the authors found that infected engorged ticks collected from hedgehogs (*Erinaceus roumanicus*) had the highest rate of mixed infection (74%) among the wildlife. The most common pathogen combination was observed to be *A. phagocytophilum*-*R. helvetica* ([28]; Appendix A). In southern Sweden, bank voles (*Myodes glareolus*) co-infected with *Candidatus N. mikurensis* and *B. afzelii* (46%) were significantly more frequent by chance than expected from the prevalence of each pathogen [60]. A 9.3% prevalence of co-infection with *B. miyamotoi* and *B. burgdorferi* s.l. was reported in ear tissues of *A. flavicollis* from Slovakia [61]. In Italy, *Apodemus* spp. ear biopsies were co-infected with *B. burgdorferi* s.l. and SFG rickettsiae [62]. Co-infection for both *Ehrlichia* and *Anaplasma* spp. with *Babesia* spp. was observed in six South African wild rodents by Reverse Line Blot (RLB) hybridization assay [82]. 

### 3.2. Meso-Mammal Species

In the northeastern US, ticks were tested for *B. burgdorferi*, *Ba. microti*, and *A. phagocytophilum* after a blood meal from opossums (*Didelphis virginiana*) and raccoons (*Procyon lotor*). The fed-ticks had a very low prevalence of co-infections as compared to ticks that fed on small rodents. The highest co-infection rate in the study was observed for *A. phagocytophilum* and *Ba. microti* (0.9%) [70]. In Canada, fourteen ticks that fed on eastern cottontail rabbits (*Sylvilagus floridanus*) were tested for different species of *Babesia*-*Borrelia* spp., and only one tick (*Haemaphysalis leporispalustris*) was found to be positive for both pathogens [72]. A low prevalence of co-infections was also observed in European grey wolves (*Canis lupus*) from Germany [63]. The authors of that study tested grey wolves for common tick-borne protozoa (*Babesia*, *Theileria*, *Hepatozoon*) and bacteria (*Anaplasma*, *Ehrlichia*, *Neoehrlichia*, *Rickettsia*) and observed that only 0.7% wolves were co-infected, all of them positive for both *A. phagocytophilum* and *H. canis*. On the other hand, high co-infection rate (17.6%) was observed in red foxes (*Vulpes vulpes*) from Poland that were positive for both tick-borne protozoans *Babesia* spp. and *H. canis* [64].

### 3.3. Large Mammal Species

Among large mammalian species, wild ungulates are considered the most important hosts due to their large size, which can harbor a large population of ticks (Figure 2). Some of these animals, such as deer, also come in proximity to domestic animals and humans while inhabiting backyards and parks in urban areas, raising human and veterinary exposure and health concerns [85,86]. White-tailed deer (*Odocoileus virginianus*) serve as an important host for *I. scapularis* (deer tick), which then contributes to the transmission of *A. phagocytophilum*, *Ba. microti*, *B. burgdorferi* and *B. miyamotoi* throughout the United States [71,83,84]. In central Maryland, *I. scapularis* ticks collected from white-tailed deer were found to have an overall co-infection rate of 25.7%. The most frequent co-infection found in these ticks was *B. burgdorferi*-*A. phagocytophilum* with 9.9% prevalence [71]. In Italy, the presence and co-infection of *A. phagocytophilum*-*Babesia* spp. were investigated in blood samples obtained from wild roe deer (*Capreolus capreolus*). Polymerase chain reaction (PCR) screening revealed the presence of at least one pathogen in 86% of the animals, while co-infection was present in approximately 19% of the tested animals [68]. In Germany, *A. phagocytophilum*-*Babesia* spp. co-infections were found in 88.4% of blood or spleen samples from roe deer; however, when engorged *I. ricinus* ticks were tested, only 6.3% of them were positive for the same combination of TBPs, and triple infections with *A. phagocytophilum*-*Babesia* spp.-*Rickettsia* spp. were reported in up to 1.5% of engorged ticks [66]. In central Spain, ticks collected from wild boar (*Sus scrofa*) and Iberian red deer (*Cervus elaphus hispanicus*) were co-infected with *Ehrlichia* spp. and/or *A. phagocytophilum* and *A. marginale* [73]. Co-infections were also reported among other ungulates such as *Bison* spp. in Mexico [65]. In that study, DNA sequencing of purified amplicons from American bison (*Bison bison*) was tested for *B. burgdorferi* s.l., *R. rickettsia*, *Ba. bovis*, *Ba. bigemina* and *A. marginale*. Out of twenty-six bison tested, nine were found to be positive for at least one pathogen, whereas two were co-infected with *B. burgdorferi* and *Ba. bovis*.

### 3.4. Birds

Birds, mainly passerines, can be fed by the Ixodid tick, which often carries TBPs, including the species of *Borrelia*, *Babesia*, *Anaplasma*, *Rickettsia*/*Coxiella*, and to some extent TBEV. The prevalence of ticks on birds depends mainly on how often certain birds feed on the ground; however, the infection rates can change over the years depending on season, geographical regions, and habitat. Due to their migratory behavior, birds can potentially transport infected ticks and easily transmit the acquired TBPs to hosts in a distant region and to new populations of healthy ticks [87]. In addition to mammals, birds also carry multiple TBPs and can facilitate their co-transmission to other hosts. In North America, *B. burgdorferi* occurrence in birds is very common. In New York, about 50% I. scapularis collected from three species of passerines (*Catharus fuscescens*, *Hylocichla mustelina* and *Turdus migratorius*) was found to be infected with *B. burgdorferi*; however, only 3% of them carried more than one TBP [70]. Conversely, in Romania, more than 30% of ticks that were collected from birds showed co-infection. The most common combinations were reported to be with *A. phagocytophilum*-*Hepatozoon* spp., *N. mikurensis*-*R. helvetica* and *Borrelia valaisiana*-*Rickettsia felis* in fed ticks [28]. When tissue samples from birds were analyzed, the existence of *A. phagocytophilum* with *Rickettsia* spp. was found to be the most common, while co-infection by three pathogens (*A. phagocytophilum*-*R. helvetica*-*B. afzelii*) was only detecte d in tissues from great tit (*Parus major*). Unfortunately, the number of samples was too small to determine overall TBCI prevalence. 

## 4. TBPs Prevalence in Reservoir Wildlife versus Surveillance of Ticks

In Germany, multiple TBPs were found in blood or spleen samples of roe deer, with *A. phagocytophilum*-*Babesia* spp. simultaneous infection as the most common, with a prevalence rate of 88.4%; however, when engorged ticks were evaluated, only 6.3% were infected with the same TBPs combination. The questing adult ticks presented a very low co-infection rate (no more than 2.5%), showing no correlation of co-infection prevalence between host and engorged or questing ticks [66]. 

Simultaneous existence of *Ba. microti* and *B. burgdorferi* was found in both questing nymphs and host-collected *I. scapularis* ticks fed on small rodents but not in ticks derived from sciurids, meso-mammals and birds [70]. In Maryland, examination of the relationship among TBPs, white-tailed deer and *I. scapularis* showed that the most common co-infection in ticks feeding on white-tailed deer was *B. burgdorferi*-*A. phagocytophilum* (9.9%) while *B. burgdorferi*-*Ba. microti* co-infections were highest in questing ticks (4.2%). Furthermore, co-infected questing ticks tended to have *B. burgdorferi*-related infections with either *Ba. microti* or *B. miyamotoi*; however, once adult females had fed on the vertebrate host, *B. burgdorferi* infections appeared to have a higher affinity for *A. phagocytophilum* [71]. Considering surveillance reports overall, it is important to pay attention to different variables such as tick sex, life stage, and feeding status before performing any correlation study between the vector and vertebrate hosts infections. In addition, the persistence of TBPs within the ticks can vary depending upon the developmental stage of ticks and on the strength of interactions among the co-infecting pathogens [71,88]. 

## 5. Major Virulence Factors of Pathogens That Cause Co-Infection

To develop promising vaccines and potential antimicrobial compounds, it is necessary to delineate key virulence factors [89]. Pathogenicity is a complex phenomenon, and multiple virulence factors affect growth and multiplication of microbes, tissue colonization by pathogens, and resulting disease. In addition to genetics, biochemical and immunological “omics” approaches are now widely used to identify functionally essential proteins of TBPs [90]. Monoclonal antibodies (MAb) and recombinant proteins are often utilized in biochemical assays to determine their critical function and contribution to microbial virulence. For example, MAbs have been raised against polymorphic immunodominant molecule (PIM), p67, and SPAG1 of *Theileria parva* to determine the role of these proteins in sporozoite entry into mammalian lymphocytes [91,92,93]. Furthermore, of particular importance in virulence factor determination are the gel-based proteomics approaches that resolve post-translation protein modifications. For instance, tyrosine phosphorylation in *A. phagocytophilum* proteins and AnkA protein as a translocated virulence factor were found to be important for pathogenesis of this organism [94]. 

Many outer surface proteins (Osps) are known to be important virulence factors of *B. burgdorferi* [95]. These lipoproteins, such as OspA, OspB, OspC, OspD, OspE, OspF, DbpA, DbpB, BBA64, BBK32, CspA, VlsE, and BptA are anchored into the outer membrane with their lipid moieties. Other proteins, such as P66, P13, Lmp1, BesC, BamA, BB0405, and Bgp are also associated with outer membrane proteins [96]. The role of a number of these virulence factors is to bind to the host receptors and extracellular matrix components (ECM) to facilitate adherence of this extracellular pathogen to colonize various cells and tissues of the host or vector. For example, OspA facilitates colonization in the tick midgut by binding to TROSPA (tick receptor for OspA) [97,98]. Similarly, OspC, which is essential for infection of mammalian hosts, binds to plasminogen in mammalian cells [99,100]. Virulence-associated P13 and P66 are putative porins and adhesins of *B. burgdorferi*, whereas VlsE presents on the surface, exhibits antigenic variation, and thereby facilitates immune evasion [101,102,103], and Bgp, DbpA, DbpB and BBK32 adhere to ECM components [104,105,106,107,108,109]. In *Anaplasma* and *Ehrlichia* spp., components of the type IV secretion system in outer membrane fractions are known to be immunogenic and important for virulence [110,111]. During bovine babesiosis caused by *Ba. bovis*, variant erythrocyte surface antigen (VESA) is transported to the infected RBCs, which promotes cytoadherence and obstructs host immune response [112,113]. Furthermore, merozoite surface antigen 2 (MSA-2) is required for host RBC invasion and is transported toward the RBC membrane [114]. A similar antigen of Ba. microti merozoites (BMSA), when used in recombinant form, has been shown to induce immune response capable of protecting experimental animals from infection [115]. Additionally, in *Babesia* spp., apical membrane antigen 1 (AMA1) and rhoptry neck protein 2 (RON2) are secreted by specialized organelles to aid in host cell invasion [116]. There have been no reports to show virulence factor association specifically with co-infections. Thus, future studies emphasizing the potential virulence factors that enhance or inhibit co-infections could lead to better understanding of pathogenesis during TBCIs. 

## 6. Animal Models for Investigation of TBD and TBCI

Animal models have greatly improved our understanding of the pathogenesis of TBDs, as infection is possible in a variety of laboratory animals, including mice, hamsters, guinea pigs, rabbits, and larger animal models such as dogs, ungulates, and nonhuman primates (NHPs). 

Several studies showed that the severity of Lyme arthritis was influenced by the *Mus musculus* genotype, with C3H/HeN or C3H/HeJ and SWR mice displaying the most severe disease [117]. Moreover, the severity of arthritis was age-dependent, as seen in C3H/HeJ, SWR, C57BL/6, SJL, and BALB/c mice, because infection of three-day-old mice showed polyarthritis after 30 days of intraperitoneal inoculation of *B. burgdorferi* [117]. Although three-week-old C3H/HeJ and SWR mice showed severe arthritis, BALB/c and C57BL/6 mice showed mild and no arthritic manifestations, respectively [117]. Arthritis severity in BALB/c mice was also dependent on the dose of inoculum used [118]. Even the severe combined immunodeficient (SCID) C57BL/6 mice develop only mild arthritis, while the C3H-SCID-strain develops progressive severe arthritis, myositis, and carditis [119]. The SCID and CD-1 mice infected with *B. miyamotoi* were used for *I. scapularis* tick feeding to then evaluate horizontal and transovarial transmission efficiency by infected ticks [120,121,122]. The DBA/1 mouse strain was found to be comparable to the C3H model for the study of experimental Lyme arthritis, which allowed for direct comparison with collagen-induced arthritis [123]. Recently, *B. burgdorferi* has also been shown to colonize the meninges, inducing inflammation of the central nervous system [124]. 

Other animals have been found to be susceptible to *B. burgdorferi*, including guinea pigs. Syrian hamsters with suppressed immune systems developed arthritis after injection of *B. burgdorferi*, but it was dependent on gamma radiation dose given [125]. The persistence of infection with *B. burgdorferi* in the heart has made it a useful model for investigations involving the pathogenesis of LD cardiac manifestations [126]. The rabbit model has provided unique opportunities to study events in the pathogenesis of Erythema migrans (EM), persistent skin infection, and visceral dissemination of Lyme borreliosis and identification of potential virulence proteins [127]. Non-human primates (NHP), such as *Rhesus macaques*, have been valuable for diagnostic assays testing, assessment of the efficacy of novel treatments, and study of persistent LD because these are the only animal models that mimic all signs of the disease, including neuroborreliosis, and they are the animals closest to humans [128,129,130,131,132]. 

The infection studies were conducted to determine *Ba. microti* infection in susceptible DBA/2 and C.B-17-SCID strains of mice. Both the susceptible DBA/2 and C.B-17.scid mice strains showed a delay in the timing of the appearance of reticulocytosis compared to C.B-17, BALB/cBy, B10.D2 strains [133]. *Ba. microti* parasitemia has been related to both the age and strains of mice. For example, young DBA/2 mice exhibit significantly higher parasitemia contrary to young C57BL/6, B10.D2 and BALB/c mice. Moreover, as age advances, both acute and persistent parasitemia increases only in DBA/2 mice [134]. Both baboons (*Papio cynocephalus*) and macaque monkeys (*Macaca mulatta*) are also susceptible to *Ba. microti* through experimental infections and transfusion-associated transmission and are invaluable for enhancing our understanding of the dynamics of parasitemia, other manifestations such as anemia and splenomegaly, and evaluation of immune responses since this model is physiologically closer to humans [135,136,137,138].

Farm animals have been used to study *A. phagocytophilum* due to their susceptibility to infection, such as sheep allowing *I. scapularis* ticks to complete the infection cycle [139] with differential expression of genes involved in the immune response in blood and tick feeding sites [140]. Recently, C3H/HeN and BALB/c mice strains as well as SCID mice showed considerable variation in the dynamics of infection by the *A. phagocytophilum* NY-18 strain [141]. C57Bl/6 male mice develop higher peripheral blood bacterial burdens and splenomegaly than female mice after infection, which is consistent with the higher incidence rate of human granulocytic anaplasmosis (HGA) in male patients [142]. 

Infection of different murine models by a human-derived *Ehrlichia* strain causing human granulocytic ehrlichiosis (HGE) demonstrated that while DBA and SCID mice remain persistently infected and C3H mice develop anemia and leukopenia [143,144], serologic cross-reactions with *B. burgdorferi* antigens were not observed [145]. Tick cell-derived inoculum for dogs and deer developed mild to no clinical signs, unlike natural tick transmission and host-specific differences in IgG responses [146].

*Bartonella* spp. can induce persistent infection in domestic and wild animals that can be a source of accidental infection in humans. Unfortunately, experimental studies in the natural host (cats) are challenging, as they can differ in the *Bartonella* strain used or the route of transmission [147,148]. Nevertheless, infection in murine models have been used with the purpose on understanding factors involved in *Bartonella* infection such as genetic susceptibility, parasitemia persistence, immunocompetence, and vertical transmission, while mother to kitten transmission is not shown in cats [149,150,151]. 

TBPs modulate vertebrate host immunity either in a manner similar or complementary to what is observed with respect to the host immunological response induced by ticks, which can affect pathogen transmission and persistence [152]. Studies using susceptible C3H mice co-infected with *Ba. microti*, *B. burgdorferi* and *A. phagocytophilum* combinations have provided significant information about the TBCI impact on vertebrate host immune response affecting respective disease manifestations [79,153]. 

Co-infection by *B. burgdorferi*-*Ba. microti* in C3H mice result in a significant increase in spirochetes burden in different organs and persistence and exacerbation of inflammatory LD manifestations because of immunosuppression caused by *Ba. microti* infection [154,155]. In addition, co-infected C3H mice also develop more pronounced inflammatory arthritis when compared to singly infected mice [156]. On the other hand, *B. burgdorferi* infection significantly attenuates *Ba. microti* parasitemia level likely because of stimulation of innate immune response by *B. burgdorferi* virulence factors, including TLR-stimulated innate immune response [154,156]. Somewhat surprising and contradictory results were reported in a preliminary study where each pathogen was shown to take independent pathways in the co-infected C3H/HeN mice; severity of arthritis, spleen weights, and parasitemia were comparable to that observed in singly infected mice [157]. Both co-infected BALB/c and C3H/HeJ mice showed similar severity of carditis [158]. C3H/HeJ mice singly infected with *Ba. microti* display slightly lower hemoglobin levels (probably because of higher parasitemia) than co-infected mice, which is compatible with attenuation of babesiosis symptoms reported during human co-infections [159]. Co-infected young C3H mice had increased IFN-γ- and TNF-α-producing T cells, a high Tregs/Th17 ratio, and diminished pathogen-specific antibody production [155]. *Ba. microti* also subverted the splenic immune response, and a marked decrease in splenic B and T cells occurred together with reduction in antibody levels and diminished functional humoral immunity in co-infected C3H mice [154]. In addition, inflammatory Lyme arthritis and spirochetes burden were reported to be more pronounced in C3H/HeJ (non-functional TLR4) than C3H/HeN (with functional TLR4) mice; signaling through TLR2, and to a lesser extent by TLR4, by *B. burgdorferi* was suggested to play an important role in determining the severity of LD manifestations [156]. 

*A. phagocytophilum* is an obligate intracellular pathogen that infects leukocytes. Several studies have shown that *B. burgdorferi*-*A. phagocytophilum* co-infection may modulate the immune response and affect the development of Lyme arthritis. Co-infected C3H mice were observed to enhance the expansion of splenic B and T cells, particularly CD4 lymphocytes, while decreasing CD8 T cells [160], together with an elevated IL-4 and reduction in IFN-γ and IL-2 cytokine levels. Moreover, co-infection increased the load of both pathogens and enhanced Lyme arthritis severity in mice compared with singly infected mice [51]. Co-infection of mice showed diminished IL-12, IFN-γ, and TNF-α levels and increased IL-6 production in addition to suppressing macrophage activation to become phagocytic. In another study, Lyme spirochete burden was found to increase in the ears, heart, and skin of co-infected C3H/HeN mice, but *Anaplasma* load was not altered [52]. Co-infected mice also had a decreased specific antibody response to *A. phagocytophilum*, but not to *B. burgdorferi* infection. Overall, the valuable insights related to bacterial virulence, disease severity, the efficiency of transmission of infection, and stimulation of immune responses using animal models are helpful to better understand the pathogenesis of different TBIs; however, a single animal model usually does not mimic all clinical features of TBDs and co-infections in humans.

## 7. Evaluation of Vaccine Candidates for Major Pathogens

Prior to the development of vaccines, other approaches were implemented to control the increasing number of TBDs. They included a broad set of efforts to minimize infection rates through chemical acaricides and repellents and through spreading education for good practice and management of disease exposure [161]. In recent years, vaccination has proven to be a better and cost-effective method to manage and control TBIs. 

Vector antigens have been known to be somewhat protective and reduce infection and transmission of pathogens in addition to diminishing tick feeding and reproduction [162]. For example, vaccines derived from the gut tissue of Boophilus microplus were shown to protect cattle from cattle tick infestations [163]. Similarly, red deer (*Cervus elaphus*) and white-tailed deer (*Odocoileus virginianus*) showed reduced *B. microplus* infestation upon administering *B. microplus*-derived recombinant BM86 or tick subolesin [164]. Subsequently, in the late 1990s, Gavac™, the vaccine formulated from recombinant BM86, was licensed, and made commercially available by Heber Biotec S.A., Havana, Cuba, in multiple Latin and Central American countries [165]. For cattle, most of the currently available vaccines to overcome TBDs are blood-attenuated and blood-derived live vaccines. These live vaccines include vaccines for theileriosis, babesiosis, anaplasmosis, and heartwater (cowdriosis) (reviewed in [90]). A reservoir-targeted vaccine for the wild white-footed mouse (*P. leucopus*) was also developed against *B. burgdorferi* s.l. Osps [166,167]. The OspA protein was found to be promising for vaccination [168]. 

Anti-tick vaccines based on mRNA have been shown to obstruct pathogen transmission to hosts and could be used together with pathogen-based vaccines for maximum protection. Recently, Sajid et al. developed a nucleoside-modified mRNA coding for 19 *I. scapularis* salivary protein (19ISP) enclosed in lipid nanoparticles [169]. Lipid nanoparticles encapsulating mRNA coding for Powassan virus genes have also demonstrated induction of potent antibodies. Additionally, the antibody response was shown to be effective against other tick-borne flaviviruses [170]. For TBEV, an interesting advancement in immunization strategy is the development of virus-like particles (VLPs) embedded with RNA replicon derived from the virus [171]. 

Several factors are important to consider when designing a vaccine. An efficient vaccine design necessitates considering the TBP transmission time after a tick bite and the dynamics of vector–host interactions that dictate the type of cycle in which the ticks participate [161]. In ecological networks, wild hosts are supported by enzootic cycles, whereas domestic animals are supported by epizootic cycles [172,173]. Both infection cycles play critical roles in increasing pathogen circulation within tick–host–pathogen networks [174]. The immune response to vaccination may differ between different hosts and therefore, need a thorough examination [175].

## 8. TBCI in Farm and Companion Animals

Veterinary TBDs are significantly impactful on human life. Diseases of farm animals adversely affect the livelihood of farmers and the economy of developing countries. Among them, the most important ones are *Rickettsia* species infections (cowdriosis and anaplasmosis) and hemoparasitic diseases caused by protozoa, including *Theileria* (theileriosis) and *Babesia* (babesiosis) [176]. In companion animals, such as cats and dogs, emerging TBDs include anaplasmosis and TBE [177]. Despite frequent reports of single infections, studies that focus on co-exposure to TBPs on domestic animals are rare. A comprehensive meta-analysis showed that approximately 3.7% of dogs and 16% of cats were co-infected with *A. phagocytophilum*-*B. burgdorferi* [25]. Furthermore, a 9.1% co-infection rate was detected in northern Europe, with nineteen different TBP combinations in *I. ricinus*, *I. persulcatus* and *Dermacentor reticulatus* ticks that were collected from dogs [21]. Co-infection with *Ba. canis* and TBEV has been reported in dogs in Poland [178]. In one study, 11% of ticks collected from domestic animals in Central China were co-infected by at least two TBPs of humans or animals [24]. Sheep, goats, and dogs may play a key role in facilitating the spread of ticks and TBDs in southern African mammal communities [179]. Ticks recovered from dogs in South Africa showed co-infections with *Ehrlichia* and *Anaplasma* spp., *A. phagocytophilum* and *Rickettsia* spp. as well as mixed *Coxiella* spp., *E. canis*, and *Rickettsia* spp. [180,181]. Bovine babesiosis and anaplasmosis are important TBDs in cattle worldwide, causing serious economic impact [182,183,184,185]. In a study from Sweden, samples from 71 domestic cattle were screened for *Babesia* and *Anaplasma* spp. with 18% of *Anaplasma*-infected animals also positive for *Ba. divergens* [183]. *Anaplasma* co-infections were also reported in cattle in Russia [186] with an observed rate of 19% co-infections between *A. marginale* and *Theileria* spp. among a total of 113 blood samples analyzed. Overall, these reports raise concerns regarding the direct economic impact and potential human risk of co-infections of TBDs from farm animals and pets.

## 9. Incidence Rate of TBCI in Humans and Impact of Co-Infections on Severity of Each Disease 

Humans can acquire multiple pathogens either by the sequential transmission of a different pathogen with more than one tick or by the transmission of multiple pathogens by a single tick [153]. The highest number of co-infections by TBPs in humans is reported in patients with LD, which is by far the most prominent TBD. The proportion of co-infection is higher in patients with relatively infrequent diseases such as babesiosis, HGA, and TBE [8]. In regions endemic for TBDs in the US, various reports have documented a 4–45% rate of either babesiosis or anaplasmosis in LD patients depending on geography [88] with infrequent existence of triple co-infection of *B. burgdorferi* with *Ba. microti*-*A. phagocytophilum* and that correlate with the co-infection prevalence of these three pathogens in ticks in the US [57]. In *Ixodes* nymphs, the prevalence of *B. burgdorferi*-*A. phagocytophilum* versus *B. burgdorferi*-*Ba. microti* is nearly the same [57], although humans have about a two-fold more incidence rate of LD with babesiosis compared to LD with anaplasmosis [88]. In European countries, the cases of co-infection of *B. burgdorferi* with *Babesia* and HGA are less common compared to LD with TBE [8,10] since the incidence of TBE is significantly higher than other TBDs in Europe [187]. 

Examination of co-infections with *B. burgdorferi* and *Ba. microti* in humans showed that both TBPs exhibited more subjective symptoms including fatigue, headache, sweats, chills, anorexia, emotional lability, nausea, conjunctivitis, and splenomegaly and longer duration of illness than those experiencing *Babesia* or *B. burgdorferi* infections alone [188]. In one case report, two patients diagnosed prepartum with LD and Ba. microti infection resulted in congenital babesiosis in both infants who developed anemia, neutropenia, and thrombocytopenia [189]. One infant required blood transfusion, supporting the need for prenatal screening of pregnant women for TBPs in the endemic regions. Infection with *A. phagocytophilum* may also be responsible for a more severe form of LD with possible long-term complications due to its effect on vascular permeability and increased production of cytokines, chemokines, and metalloproteases [190]. Moreover, in 2013, patients with EM and positive blood culture for *A. phagocytophilum* experienced longer illness duration than those with HGA alone [191]. One pathogenic infection can be expressed clinically more prominently in different TBCI while patients may remain asymptomatic for the co-infecting pathogen. In the Netherlands, 2.7% of the cases of EM with detection of *B. burgdorferi* had a probability of co-infection with *N. mikurensis*, *A. phagocytophilum*, *Ba. divergens*, or *B. miyamotoi* without additional symptoms other than those reported for Lyme borreliosis during the three-month follow-up period [192]. Recently, a systematic review of 655 cases of possible co-infections of patients with neurological symptoms linked to TBEV and neuroborreliosis were found to be the most frequent co-infections, but information about severity of clinical manifestations was scarce [8]. The most common diagnosed co-infections can generally occur between *Borrelia* and *Babesia* spp., *A. phagocytophilum*, or TBEV [10,193] and the effect could lead to an increase or decrease in the respective disease severity. One of major challenges of human studies in TBCI is distinguishing active infection from past exposure and assessing the severity frequency because of the use of indirect serological tests for diagnosis and because antibodies persist long after infections are cleared.

## 10. Vaccines against TBDs for Human Use 

Only limited vaccines are approved for TBDs in humans. The phase III trials of B. burgdorferi OspA vaccines were first reported in 1998 [194,195] and showed vaccines to be safe and effective. After OspA165–173 was demonstrated to be an immunodominant epitope for T helper cells containing partial sequence homology with a peptide of human lymphocyte function-associated antigen-1 (hLFA-1), the latter was considered a cross-reactive autoantigen and hence, associated likely with autoimmunity [196,197]. In 2002, the only human vaccine (LYMErix™), which used *B. burgdorferi* OspA to allow antibodies to target the spirochetes in the tick gut [194] and prevented migration to the new host, was voluntarily removed from the US market due to concerns of patients regarding stimulation of arthritic manifestations by the vaccine. In Europe, human trials with a modified OspA vaccine against spirochetes expressing the combination of six OspA serotypes are still ongoing [198,199]. Other novel strategies have focused on identifying proteins of TBPs involved in the infection of vertebrate host cells or microbial cell surface proteins as potential vaccine candidates.

There are currently four vaccines available for preventing TBEV infection in humans in Europe and Russia where this infection is prevalent [200]. These vaccines showed a high level of immunogenicity and protection and an excellent safety profile. In August 2021, the US Food and Drug Administration (FDA) agency approved TICOVAC™, an inactivated virus vaccine for active immunization, to prevent TBEV infection in individuals one year of age and older [201,202]. 

The use of inactivated *R. rickettsia* intact bacteria or subunit vaccines provided only limited immunity and have not been successful until now [203,204]. Human vaccines for *Anaplasma* or *Babesia* are not currently available. 

## 11. Economic Impact of TBDs

The information on the economic impact of TBDs focuses mainly on direct medical costs in humans and damage on livestock, and there is no mention of the economic burden of co-infections. An average of approximately USD 73 million was spent annually on TBDs in the US from 2006 to 2010, with major funding expenditure focused on academia, microbiological studies, and clinical research, and less than 25% of the funding spent on prevention and surveillance [205]. LD is one of the largest TBDs in the US, and most of the research and data related to financial cost focus on this disease [206]. A cost-of-illness study in the US reported an expected national expenditure included both direct medical and indirect costs of USD 4.45 billion over five years (USD 2.5 billion, US inflation calculator (USIC) 1996) for treatment interventions to prevent LD sequelae [207]. Reports from the 2000s show an inflated annual economic impact of USD 282–643 million for LD [208,209] (USIC 2006: USD 203 million; USIC 2008: USD 492 million, respectively) making it clear that LD direct medical costs and testing are costly and will continue to increase as the cases increase further. A safe and cost-effective approach to Lyme cardiac disease revealed cost savings when an early permanent pacemaker implant was used with a cost of USD 48,711 compared to a temporary pacing wire cost of USD 147,121 in 2011 [210].

In 2015, an estimated LD expense of USD 740 million annually was reported in the US, and post-treatment LD syndrome-related diagnoses were associated with marked increases in costs up to 1.3 billion per year [211]. The number of patients who remain ill over time are growing, while LD cases remaining underreported. In Europe, several countries are economically affected by LD [212,213,214,215,216]. As a result, Germany has been strongly investing in LD diagnostics, spending up to EUR 51.2 million as reported in 2008 [216]. 

For non-LD TBDs, diagnostic costs in 2008 in USD were 5.5 million for Rocky Mountain spotted fever, 3.7 million for ehrlichiosis, 2 million for babesiosis, 1.6 million for anaplasmosis, 20,000 for tick-borne relapsing fever and 5400 for Colorado tick fever, with a total estimated direct cost of 12.9 million (USD 9.6 million, USIC 2008) in the United States [217]. In addition, the cost-effectiveness of blood donation screening techniques showed an incremental cost-effectiveness ratio of USD 54,000 to 83,000, quality-adjusted life-year compared to each test alone in the US [218]. In comparison, for hospitalization involving human babesiosis cases, an estimated median cost of EUR 4195.62 (EUR 2017) was incurred in Spain [219]. These calculations provide true magnitude and insight for effective TBDs diagnostic test utilization and surveillance costs. Nevertheless, more comprehensive and inclusive investigations conducted in the affected areas are necessary to accurately determine the actual economic impact of TBDs.

The damage to livestock due to TBDs has serious economic impact, especially in developing countries, reducing livestock production, and causing mortality [220]. Among the most economically significant veterinary TBDs are babesiosis, anaplasmosis, theileriosis, and cowdriosis [185]. A study in Tanzania found an annual loss of USD 364 million incurred nationally from the losses of production, treatment, and control of TBDs, out of which 68% was attributed only to theileriosis [221]. In India, the annual economic loss estimates were as high as USD 498.7 million [222]. In Australia, the pathogenic strain *T. orientalis* Ikeda was first identified in 2011 [223], and by 2014, had affected 25% of the cattle, resulting in massive losses to the milk and beef industries [224]; however, theileriosis has historically been caused by benign and rare strains in the US. The situation is on the verge of changing, as *T. orientalis* Ikeda has been recently identified and is competent with *H. longicornis* ticks in the US [225]. The estimated global cost of TBDs on cattle borne annually is from USD 13.9 to 18.7 billion [226]. 

## 12. Summary

In this review, we have summarized studies documenting the prevalence of TBDs and TBCI in different reservoir animals and their correlation with infected tick surveillance studies. The persistence of co-infecting pathogens within tick vectors can vary depending upon the developmental stage, and the strength of the interactions among the co-existing pathogens was reported. Therefore, consideration of different variables such as tick sex, life stage, and feeding status is needed before performing any correlation study among different hosts. Based on the literature reviewed, we have consistently observed that the profile of co-infecting pathogens changes according to the region studied and the type of samples analyzed, e.g., engorged versus questing ticks, or animal tissues. Different host species can also harbor different pathogen combinations in different regions. For example, in the US, the white-footed mice and meadow voles have high co-infection rates, while in Europe, the highest co-infection prevalence has been observed in Apodemus and hedgehogs (Figure 2). *B. burgdorferi*-*Ba. microti* and *B. burgdorferi*-*A. phagocytophilum* co-infections occur more frequently in the US, whereas *Babesia*-*A. phagocytophilum* and *Rickettsia*-*A. phagocytophilum* are more frequently observed in Europe. Co-infection rates of ungulates, especially deer, were found to be similar in both the US and Europe (Figure 1). 

We have highlighted major virulence factors identified in TBPs using animal models, as well as the impact of co-infections on host immune response and disease severity. Additionally, our emphasis on animal models brings valuable insights into the recent advances and limitations of investigations of TBDs. Studies on small rodents have helped to evaluate factors such as age, gender, and genetic background susceptibility to the outcome of each disease and the possible roles of immunopathogenesis in disease severity, identification and testing of the efficacy of tick-vaccine candidates, etc. Evaluation of vaccine candidates and the economic impact of these infections are also briefly described. Despite significant successes, efforts should be made to fully appreciate the outcomes of co-infections by the use and refinement of animal model systems. This will lead to greater knowledge and fill the gap in the understanding of TBCI. 

## Figures and Tables

**Figure 1 pathogens-11-01309-f001:**
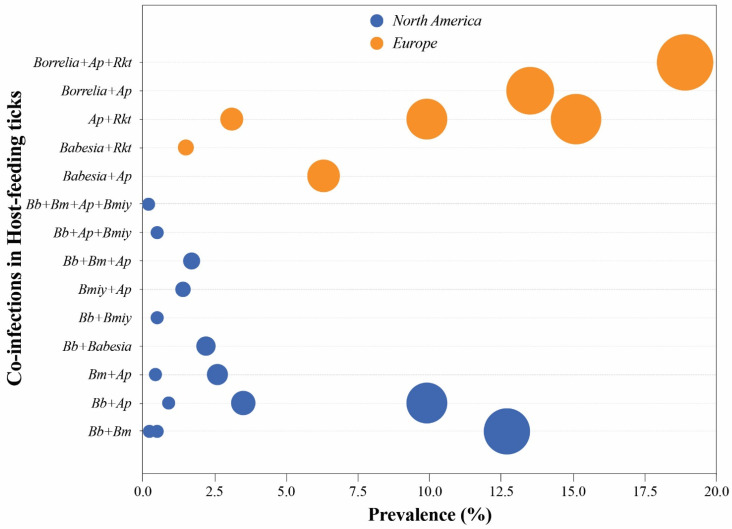
**Most common tick-borne co-infections in host-feeding ticks from two regions: North America and Europe.** Each bubble represents one data point where the size indicates the prevalence, and the color represents the geographic area. Data were collected from different sources listed in Table 1. Abbreviations used in the figure: *Bb*, *Borrelia burgdorferi*; *Bmiy*, *Borrelia miyamotoi*; *Bm*, *Babesia microti*; *Ap*, *Anaplasma phagocytophilum*; *Rkt*, *Rickettsia* spp.

**Figure 2 pathogens-11-01309-f002:**
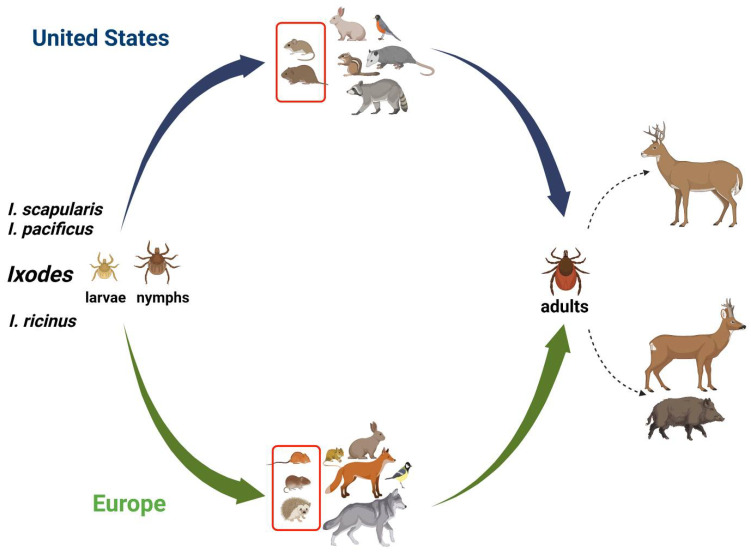
**The schematic transmission cycle of TBIs in wildlife.** Transmission to a wide range of vertebrate hosts including mammals and birds occurs by tick bites during a blood meal. Tick larvae and nymphs feed mainly on small rodents. In the United States, the white-footed mice and meadow voles are reservoir hosts for TBIs, while in Europe, they vary from the striped field mouse and bank voles to hedgehogs. Adult female ticks feed primarily on deer, such as white-tailed deer in the US and roe deer and wild boar in Europe. The red rectangle in host vertebrates represents a high prevalence of TBPs as described in the main text and Table 1. Solid arrows represent the transmission cycle, blue for the United States and green for Europe. Dashed arrows represent tick feeding. This figure was created using BioRender.com and summarizes previously published data on hosts in the wild [15,28,33,34,57,60,61,63,68,70,71,72,73,83,84].

**Table 1 pathogens-11-01309-t001:** TBCI prevalence in wildlife and ticks.

Host	Region	Host Species	Co-Infections	Prevalence (%)	Ref.
Small rodents	US	*P. leucopus*; *M. pennsylvanicus*	*Bm* + *Bb*	35.7	[15]
		*P. leucopus*		9.4	[58]
				12	[59]
			*Bm* + *Ap*	<1	[59]
				1	[58]
			*Bb* + *Ap*	22.5	[58]
				1	[59]
			*Bb* + *Ech*	2	[59]
			*Ap* + *Ech*	<1	[59]
			*Bm* + *Bb* + *Ap*	12.8	[58]
				6	[59]
			*Bm* + *Bb* + *Ech*	2	[59]
			*Bb* + *Ap* + *Ech*	<1	[59]
			*Bm* + *Bb* + *Ap* + *Ech*	2	[59]
	Europe	*Myodes glareolus*	*Ba* + *CNm*	42	[60]
		*Apodemus flavicollis*	*Bb* + *Bmiy*	9.3	[61]
		*Apodemus* spp.	*Rs* + *Bl/Bv*	2	[62]
Meso mammals	Europe	*Canis lupus* (grey woof)	*Ap* + *Hc*	<1	[63]
		*Vulpes vulpes* (red fox)	*Babesia* spp. + *Hc*	17.6	[64]
Ungulates (Large mammals)	Mexico	*Bison bison* (American Bison)	*Bbov* + *Bb*	7.6	[65]
	Europe	*Capreolus capreolus* (roe deer)	*Babesia* spp. + *Ap*	88.4	[66]
				79.9	[67]
				18.92	[68]
		*Cervus elaphus* (red deer)		1.9	[69]
				62.2	[67]
**Vector**		**Vector species**			
Questing ticks	US	*I. scapularis*	*Bm* + *Bb*	6.68	[70]
				4.2	[71]
			*Bb* + *Ap*	2.47	[70]
				1.8	[71]
			*Bm* + *Ap*	<1	[70]
			*Bm* + *Bmiy*	<1	[71]
			*Bb* + *Bmiy*	1.5	[71]
			*Bm* + *Bb* + *Ap*	<1	[71]
			*Bm* + *Bb* + *Bmiy*	<1	[71]
	Canada	*I. scapularis*	*Babesia* spp. + *Bb*	1.15	[72]
	Europe	*I. ricinus*	*Babesia* spp. + *Ap*	<1	[66]
			*Babesia* spp. + *Rickettsia* spp.	1.5	[66]
			*Ap* + *Rickettsia* spp.	1.5	[66]
Engorged ticks (Collected from birds, rodents, meso mammals and ungulates)	US	*I. scapularis*	*Bm* + *Bb*	<1, 12.69	[70]
				<1	[71]
			*Bm* + *Ap*	<1	[70]
				2.6	[71]
			*Bb* + *Ap*	9.9	[71]
				<1, 3.5	[70]
			*Bb* + *Bmiy*	<1	[71]
			*Bmiy* + *Ap*	1.4	[71]
			*Bm* + *Bb* + *Ap*	1.7	[71]
			*Bb* + *Ap* + *Bmiy*	<1	[71]
			*Bm* + *Bb* + *Bmiy* + *Ap*	<1	[71]
	Canada	*Ixodes* spp. and *Haemaphysalis leporispalustris*	*Babesia* spp. + *Bb*	2.2	[72]
	Europe	*I. ricinus*	*Babesia* spp. + *Ap*	6.3	[66]
			*Babesia* spp. + *Rickettsia* spp.	1.5	[66]
			*Ap* + *Rickettsia* spp.	15.1	[66]
			*Ap* + *Theileria* spp.	<1	[28]
			*Ap* + *Hepatozoon* spp.	<1	[28]
			*Borrelia* spp. + *Ap* + *Rh* + *Theileria* spp.	<1	[28]
			*Bbav/Bg* + *Ap*	<1	[28]
			*Ba/Bv* + *Rickettsia* spp.	<1	[28]
			*Bmiy* + *Ap*	<1	[28]
		*Ixodes* spp.	*Ap + Rh*	9.9	[28]
			*Bg* + *Ap*	3.1	[28]
			*Borrelia* spp. + *Ap* + *Rickettsia* spp.	18.9	[28]
			*Bbav/Bg* + *Ap* + *Nm*	<1	[28]
		*Ixodes* spp. and *Haemaphysalis*	*Ba* + *Ap*	9	[28]
		*Punctata*	*Ap* + *Rm*	3.1	[28]
		*Hyalomma marginatum*	Prp + *Ehrlichia* spp.	12.19	[73]
		*marginatum*, *Rhipicephalus bursa*	*Ehrlichia* spp. + *Ap*	9.75, 12.76	[73]
		and *Dermacentor marginatus*	*Ehrlichia* spp. + *Am*	4.88	[73]
			*Ehrlichia* spp. + *Ap* + *Am*	4.88, 6.38	[73]
			Prp + *Ehrlichia* spp. + *Ap* + *Am*	7.32	[73]

Abbreviations: *Bb*, *Borrelia burgdorferi*; *Bmiy*, *Borrelia miyamotoi*; *Bg*, *Borrelia garinii*; *Bv*, *Borrelia valaisiana*; *Ba*, *Borrelia afzelii*; *Bbav*, *Borrelia bavariensis*; *Bl*, *Borrelia lusitaniae*; *Bm*, *Babesia microti*; *Bd*, *Babesia divergens*; *Bbov*, *Babesia bovis*; *Ap*, *Anaplasma phagocytophilum*; *Am*, *Anaplasma marginale*; *Rh*, *Rickettsia helvetica*; *Rm*, *Rickettsia monacensis*; *Rs*, *Rickettsia slovaca*; *Ech*, *Ehrlichia chaffeensis*; *Hc*, *Hepatozoon canis*; *Nm*, *Neoehrlichia mikurensis*; *CNm*, *Candidatus Neoehrlichia mikurensis*; Prp, Piroplasma. See Appendix A: data table adapted from Borsan et al., 2021 [28].

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
