# Peer review of "Transmission Cycle of Tick-Borne Infections and Co-Infections, Animal Models and Diseases"

_pathogens, 2022, doi:10.3390/pathogens11111309_

Round 1

Reviewer 1 Report

I read with interest paper authored by Sandra C. Rocha et al. The ms is worth publishing and consist wide range of information about tick-borne infections, co-infections, prevalence of TBD and animal models and diseases.

I find some flaws that should be revised:

L 42 - . instead of >

L 49 – double space

L 108 - : instead of ;

Table 1 – I suggest to revised Table 1 (and connected with that the references list) in term of the host species like small rodents in Europe e.g. Myodes glareolus or Apodemus flavicolis which are very common host species and co-infections of TBP in this rodents are also known (e.g. co-infections of Borrelia afzelii and Candidatus Neorhrlichia mikurenisis in Sweden or B. miyamotoi and B.burgdorferii in Slovakia). Also large mammals which are very common as host for ticks in Europe e.g. Sus scrofa are often co-infected by tick-borne pathogens.

Examples of literature below:

Andersson, M., Scherman, K., & Råberg, L. (2014). Infection dynamics of the tick-borne pathogen “Candidatus Neoehrlichia mikurensis” and coinfections with Borrelia afzelii in bank voles in Southern Sweden. Applied and environmental microbiology, 80(5), 1645-1649.

Hamšíková, Z., Coipan, C., Mahríková, L., Minichová, L., Sprong, H., & Kazimírová, M. (2017). Borrelia miyamotoi and co-infection with Borrelia afzelii in Ixodes ricinus ticks and rodents from Slovakia. Microbial ecology, 73(4), 1000-1008.

de la Fuente, J., Naranjo, V., Ruiz-Fons, F., Vicente, J., Estrada-Peña, A., Almazán, C., ... & Gortázar, C. (2004). Prevalence of tick-borne pathogens in ixodid ticks (Acari: Ixodidae) collected from European wild boar (Sus scrofa) and Iberian red deer (Cervus elaphus hispanicus) in central Spain. European Journal of Wildlife Research, 50(4), 187-196.

Or change the Table title for e.g. „Choosen TBCI prevalence….”

I think it will be also interestnig add birds as a host to the Table 1 too.

Table 1 – I. scapularis instead of I. Scapularis (twice – in section vectors and engorged ticks) and I. ricinus instead of I. Ricinus (section engorged ticks – Canada)

L 188 – Latin name of Western Chipmunk

L 320 – apical instead of Apical

L 322 – double space

L 359 – Rhesus macaques instead of Rhesus macaques

L 370 – 371 – Latin name of both monkeys

L 496 – I. ricinus instead of I. Ricinus

L 497 – Dermacentor reticulatus is given here for the first time and should be written without abberviation

Section 10 – I would recommended to add information about new perspectives in vaccines based on the principle of DNA and mRNA against Lyme boreliosis which represent one of the innovative and alternative approaches as standard research and are on the top now.

Reviewer 2 Report

This paper reviewed transmission cycle of tick-borne infections, animal models and diseases. Reading the manuscript it is evident that most of the text focus on tick-borne co-infections (TBCI) from different perspectives/aspects.

General comment:

The manuscript is well written (only some typing errors, check for double space), sub-chapters are complete and exhaustive. Figure and tables are complete and easy to understand.

Many aspects of tick-borne infections (tick-borne pathogens, transmission cycles, animal models and tick-borne diseases) have been covered through the text and therefore, the text is very long and should consider to shorten it.

In my opinion, authors should focus only on TBCI adding different information than Gomez-Chamorro et al. 2021 and Culter et al. 2021.

For example: chapter 6.1 could be removed or briefly integrated in chapter 6.2 or in the introduction. Title should be rewritten in the agreement with the new text.

“Ecological and evolutionary perspectives on tick-borne pathogen co-infections” Gomez-Chamorro et al. 2021

“Tick-borne diseases and co-infection: Current considerations” Culter et al. 2021

In my opinion, this study deserves to be published for the value of the information collected on tick-borne co-infections with major revisions.

Specific comments:

Line 29: add “s” after “TBD”

Line 32-35 could you please rephrased this sentence splitting it

Line 35-38: are you referring to “invasion” or “spreading” or “increasing in number”? Actually, ticks are doing all these three actions. Please rephrased this sentences.

Line 42: “>”? If it is a typo please delete it

Line 45: add “Crimean-” before “Congo”

Line 46: add “s” after “TBD”

Line 47: rephrased as follow: … (U.S.); of these, the four more prevalent illnesses are represented by LD (34,945 cases), …

Line 49: double space between “that” and “<10%”

Line 49-54: could you please rephrased this paragraph

Line 56: delete “of ticks”

Line 80-81: rephrased as follow: “4) the use of animal models for infection, co-infection, and pathogenesis studies,”

Line 87: replace “in association” with “capacity to harbor”

Line 97-98: join the sentences

Line 112: ad “,” after “,the host reservoir”

Line 172: double space between “,and” and “transplacental”

Line 473 and 489: capital letter or not? In my “cowdriosis” is correct

Line 480-482: I understood what you want to say but it is confusing, please rephrased it

Line 509: replace “harm” with “Damage”

Line 509-518: this paragraph should be moved in “Economic impact of TBDs”

Line 587 and 615: double space

Line 672: Ba microti (no capital letter for specie)

Figure 2: all the tick stages are in the plural form, I suggest to replace “larva” with “larvae” for homogeneity

Supplementary material:

no capital letter for specie (Ba microti and Bo. afzelli)

Add space between genus and specie (R. helvetica)

Table should be self-explicative therefore, please, do not use abbreviation unless you add the legend in which you explain the abbreviations

Reviewer 3 Report

The review is well written for the study titled  'Transmission Cycle of Tick-borne infections, Animal Models and Diseases'. However seeing that this is a review and the title does not limit to only USA and Europe, it will be beneficial to take into consideration  other countries research outputs. 

Espinaze, M.P., Hellard, E., Horak, I.G. and Cumming, G.S., 2018. Domestic mammals facilitate tick-borne pathogen transmission networks in South African wildlife. Biological Conservation221, pp.228-236.

De Boni, L., 2017. Occurrence of tick-borne haemoparasites in selected South African wild rodent species in the Mnisi Communal area (Doctoral dissertation, University of Pretoria).

Kolo, A.O., Sibeko-Matjila, K.P., Maina, A.N., Richards, A.L., Knobel, D.L. and Matjila, P.T., 2016. Molecular detection of zoonotic rickettsiae and Anaplasma spp. in domestic dogs and their ectoparasites in Bushbuckridge, South Africa. Vector-Borne and Zoonotic Diseases16(4), pp.245-252.

Round 2

Reviewer 2 Report

The authors properly replied to each comment. There is only one final correction to be made which I report below:

Line 35: “Tick populations are expanding in terms of species types and new geographical areas …”

Ticks are growing in term of abundance (increasing in number, higher frequency of detection) and are expanding in new geographical areas. Actually, no new tick species have recently been discovered. Please rephrase the sentence

Author Response

Please find updated manuscript with revised sentence marked in blue font in the word file. Previous revisions are now reverted to black font to make it convenient for reviewer to find the change.